# PARAMETRIC EXPONENTIAL LINEAR UNIT FOR DEEP CONVOLUTIONAL NEURAL NETWORKS

**Ludovic Trottier, Philippe Giguère & Brahim Chaib-draa**
Department of Computer Science and Software Engineering
Laval University, Quebec, Canada
`ludovic.trottier.1@ulaval.ca`
`philippe.giguere, brahim.chaib-draa@ift.ulaval.ca`

## ABSTRACT

The activation function is an important component in Convolutional Neural Networks (CNNs). For instance, recent breakthroughs in Deep Learning can be attributed to the Rectified Linear Unit (ReLU). Another recently proposed activation function, the Exponential Linear Unit (ELU), has the supplementary property of reducing bias shift without explicitly centering the values at zero. In this paper, we show that learning a parameterization of ELU improves its performance. We analyzed our proposed Parametric ELU (PELU) in the context of vanishing gradients and provide a gradient-based optimization framework. We conducted several experiments on CIFAR-10/100 and ImageNet with different network architectures, such as NiN, Overfeat, All-CNN and ResNet. Our results show that our PELU has relative error improvements over ELU of 4.45% and 5.68% on CIFAR-10 and 100, and as much as 7.28% with only 0.0003% parameter increase on ImageNet. We also observed that Vgg using PELU tended to prefer activations saturating closer to zero, as in ReLU, except at the last layer, which saturated near -2. Finally, other presented results suggest that varying the shape of the activations during training along with the other parameters helps controlling vanishing gradients and bias shift, thus facilitating learning.

## 1 INTRODUCTION

Over the past few years, Convolutional Neural Networks (CNNs) have become the leading approach in computer vision (Krizhevsky et al., 2012; LeCun et al., 2015; Vinyals et al., 2015; Jaderberg et al., 2015; Ren et al., 2015; Hosang et al., 2016). Through a series of non-linear transformations, CNNs can process high-dimensional input observations into simple low-dimensional concepts. The key principle of CNNs is that features at each layer are composed of features from the layer below. This creates a hierarchical organization of increasingly abstract concepts. Since levels of organization are often seen in complex biological structures, such a hierarchical organization makes CNNs particularly well-adapted for capturing high-level abstractions from real-world observations.

The activation function plays a crucial role in learning representative features. Defined as $\max\{h, 0\}$, the Rectified Linear Unit (ReLU) is one of the most popular activation function (Nair & Hinton, 2010). It has interesting properties, such as low computational complexity, non-contracting first-order derivative and induces sparse activations, which have been shown to improve performance Krizhevsky et al. (2012). The main drawback of ReLU is its zero derivative for negative arguments. This blocks the back-propagated error signal from the layer above, which may prevent the network from reactivating dead neurons. To overcome this limitation, Leaky ReLU (LReLU) adds a positive slope $a$ to the negative part of ReLU (Maas et al., 2013). Defined as $\max\{h, 0\} + a\min\{h, 0\}$, where $a > 0$, LReLU has a non-zero derivative for negative arguments. Unlike ReLU, its parameter $a$ allows a small portion of the back-propagated error signal to pass to the layer below. By using a small enough value $a$, the network can still output sparse activations while preserving its ability to reactivate dead neurons. In order to avoid specifying by hand the slope parameter $a$, Parametric ReLU (PReLU) directly learns its value during back-propagation (He et al., 2015b). As the training phase progresses, the network can adjust its weights and biases in conjunction with the slopes $a$ of all its PReLU for

potentially learning better features. Indeed, He et al. (2015b) have empirically shown that learning the slope parameter $a$ gives better performance than manually setting it to a pre-defined value.

A recently proposed important activation function is the Exponential Linear Unit (ELU). Is is defined as identity for positive arguments and $a(\exp(h) - 1)$ for negative ones (Clevert et al., 2015). The parameter $a$ can be any positive value, but is usually set to $1$. ELU has the interesting property of reducing *bias shift*, which is defined as the change of a neuron's mean value due to weight update. If not taken into account, bias shift leads to oscillations and impeded learning (Clevert et al., 2015). Clevert et al. (2015) have shown that either centering the neuron values at zero or using activation functions with negative values can reduce bias shift. Centering the neuron values can be done with the Batch Normalization (BN) method (Ioffe & Szegedy, 2015), while adding negative values can be done with parameterizations such as LReLU or PReLU.

Based on the observation that learning a parameterization of ReLU improves performance (He et al., 2015b), we propose the Parametric ELU (PELU) that learns a parameterization of ELU. We define parameters controlling different aspects of the function and propose learning them during back-propagation. Our parameterization preserves differentiability by acting on both the positive and negative parts of the function. Differentiable activation functions usually give better parameter updates during back-propagation (LeCun et al., 2015). PELU also has the same computational complexity as ELU. Since parameters are defined layer-wise instead of per-neurons, the number of added parameters is only $2L$, where $L$ is the number of layers. Our experiments on the CIFAR-10/100 and ImageNet datasets have shown that ResNet (Shah et al., 2016), Network in Network (Lin et al., 2013), All-CNN (Springenberg et al., 2015) and Overfeat (Sermanet et al., 2013) with PELU all had better performances than with ELU. We finally show that our PELUs in the CNNs adopt different non-linear behaviors during training, which we believe helps the CNNs learning better features.

The rest of the paper is organized as follows. We present related works in section 2 and described our proposed approach in section 3. We detail our experimentations in section 4 and discuss the results in section 5. We conclude the paper in section 6.

## 2    RELATED WORK

Our proposed PELU activation function is related to other parametric approaches in the literature. The Adaptive Piecewise Linear (APL) unit learns a weighted sum of $S$ parametrized Hinge functions (Agostinelli et al., 2014). One drawback of APL is that the number of points at which the function is non-differentiable increase linearly with $S$. Moreover, though APL can be either a convex or non-convex function, the rightmost linear function is forced to have unit slope and zero bias. This may be an inappropriate constraint which could affect the representation ability of the CNNs.

Another activation function is Maxout, which outputs the maximum over $K$ affine functions for each input neuron (Goodfellow et al., 2013). The main drawback of Maxout is that it multiplies by $K$ the amount of weights to be learned in each layer. For instance, in the context of CNNs, we would apply a max operator over the feature maps of each $K$ convolutional layers. This could become too computationally demanding in cases where the CNNs are very deep. Unlike Maxout, our PELU adds only $2L$ parameters, where $L$ is the number of layers.

Finally, the S-Shaped ReLU (SReLU) imitates the Webner-Fechner law and the Stevens law by learning a combination of three linear functions (Jin et al., 2015). Although this parametric function can be either convex or non-convex, SReLU has two points at which it is non-differentiable. Unlike SReLU, our PELU is fully differentiable, since our parameterization acts on both the positive and negative sides of the function. This in turn improves the back-propagation weight and bias updates.

## 3    PARAMETRIC EXPONENTIAL LINEAR UNIT (PELU)

In this section, we present our proposed PELU function and analyze it in the context of vanishing gradients. We also elaborate on the gradient descent rules for learning the parameterization.

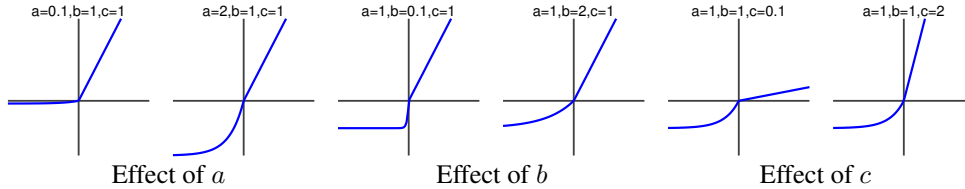

Figure 1: Effects of parameters $a$, $b$ and $c$: The saturation point decreases when $a$ increases, it saturates faster when $b$ decreases, and the slope of the linear part increases when $c$ increases.

### 3.1 DEFINITION

The standard Exponential Linear Unit (ELU) is defined as identity for positive arguments and $a(\exp(h) - 1)$ for negative arguments (Clevert et al., 2015). Although the parameter $a$ can be any positive value, Clevert et al. (2015) proposed using $a = 1$ to have a fully differentiable function. For other values $a \neq 1$, the function is non-differentiable at $h = 0$. For this reason, we do not directly learn parameter $a$ during back-propagation. Updating $a$ with the gradient would break differentiability at $h = 0$, which could imped back-propagation.

We start by adding two additional parameters to ELU as follows:

$$f(h) = \begin{cases} ch & \text{if } h \geq 0 \\ a(\exp(\frac{h}{b}) - 1) & \text{if } h < 0 \end{cases}, \quad a, b, c > 0, \tag{1}$$

for which the original ELU can be recovered when $a = b = c = 1$. As shown in Figure 1, each parameter in (1) controls different aspects of the activation. Parameter $c$ changes the slope of the linear function in the positive quadrant (the larger $c$, the steeper the slope), parameter $b$ affects the scale of the exponential decay (the larger $b$, the smaller the decay), while $a$ acts on the saturation point in the negative quadrant (the larger $a$, the lower the saturation point). We also constrain the parameters to be positive to have a monotonic function. Consequently, reducing the weight magnitude during training always lowers the neuron contribution.

Using this parameterization, the network can control its non-linear behavior throughout the course of the training phase. It may increase the slope with $c$ or the decay with $b$ to counter vanishing gradients, and push the mean activation towards zero by lowering the saturation point with $a$ for better managing bias shift. We now look into gradient descent and define update rules for each parameter, so that the network can adjust its behavior as it seems fit. However, a standard gradient update on parameters $a, b, c$ would make the function non-differentiable at $h = 0$ and impair back-propagation. Instead of relying on a projection operator to restore differentiability after each update, we constrain our parameterization by forcing $f$ to stay differentiable at $h = 0$. We equal the derivatives on both sides of zero, and solve for $c$:

$$\left.\frac{\partial ch}{\partial h}\right|_{h=0} = \left.\frac{\partial a(\exp(\frac{h}{b}) - 1)}{\partial h}\right|_{h=0} \tag{2}$$

which gives $c = \frac{a}{b}$ as solution. Incorporating (2) gives the proposed Parametric ELU (PELU):

$$\text{PELU:} \qquad f(h) = \begin{cases} \frac{a}{b}h & \text{if } h \geq 0 \\ a(\exp(\frac{h}{b}) - 1) & \text{if } h < 0 \end{cases}, \quad a, b > 0 \tag{3}$$

With this parameterization, in addition to changing the saturation point and exponential decay respectively, both $a$ and $b$ adjust the slope of the linear function in the positive part to ensure differentiability at $h = 0$.

### 3.2 ANALYSIS

To understand the effect of the proposed parameterization, we now investigate the vanishing gradient for the following simple network, containing one neuron in each of its $L$ layers:

$$x = h_0, \qquad h_l = w_l h_{l-1}, \qquad z_l = f(h_l), \qquad E = \ell(z_L, y) \qquad (1 \leq l \leq L) \tag{4}$$

where we have omitted, without loss of generality, the biases for simplicity. In (4), $\ell$ is the loss function between the network prediction $z_L$ and label $y$, which takes value $E$ at $x$. In this case, it can be shown using the chain rule of derivation that the derivative of $E$ with respect to any weight $k$ is:

$$\frac{\partial E}{\partial w_k} = h_{k-1} f'(h_k) \left[ \prod_{j=k+1}^{L} f'(h_j) w_j \right] \frac{\partial E}{\partial z_L} ,$$

where $f'(h_k)$ is a shortcut for $\frac{\partial z_k}{\partial h_k}$. Vanishing gradient happens when the product term inside the bracket has a very small magnitude, which makes $\frac{\partial E}{\partial w_k} \approx 0$. Since the updates are proportional to the gradients, the weights at lower layers converge more slowly than those at higher layers, due to the exponential decrease as $k$ gets smaller (the product has $L - k$ terms). One way the network can fight vanishing gradients is with $f'(h_j) w_j \equiv f'(w_j h_{j-1}) w_j \geq 1$, so that the magnitude of the product does not tend to zero. Therefore, a natural way to investigate vanishing gradient is by analyzing the interaction between weight $w$ and activation $h$, after dropping layer index $j$. Specifically, our goal is to find the range of values $h$ for which $f'(wh)w \geq 1$. This will indicate how precise the activations $h$ must be to manage vanishing gradients.

**Theorem 1.** *If $w \geq \frac{b}{a}$ and $h < 0$, then $w^* = \exp(1)\frac{b}{a}$ maximizes the interval length of $h$ for which $f'(wh)w \geq 1$, which length takes value $l^* = a \exp(-1)$.*

*Proof.* With our proposed PELU, we have:

$$f'(wh)w = \begin{cases} w\frac{a}{b} & \text{if } h \geq 0 \\ w\frac{a}{b}\exp(wh/b) & \text{if } h < 0 \end{cases}, \quad a, b > 0 \tag{5}$$

Assuming $w \geq \frac{b}{a}$ and $h < 0$, and using the fact that $w\frac{a}{b}\exp(wh/b)$ is monotonically increasing, the interval length $l(w)$ of values $h$ for which $w\frac{a}{b}\exp(wh/b) \geq 1$ is given by the magnitude of the zero of $w\frac{a}{b}\exp(wh/b) - 1$. Solving the derivative equals zero for $h$ gives $l(w) = |\log(\frac{b}{a}\frac{1}{w})|(\frac{b}{w})$. Using the fact that $w \geq \frac{b}{a}$, it can be shown that $l(w)$ is pseudo-concave, so it has a unique optimum. Maximizing $l(w)$ with respect to $w$ is thus the solution of solving the derivative equals zero, which gives $l^* = a \exp(-1)$, at $w^* = \exp(1)\frac{b}{a}$. $\qquad\square$

This result shows that in the optimal scenario where $w = \exp(1)\frac{b}{a}$, the length of negative values $h$ for which $f'(wh)w \geq 1$ is no more than $a \exp(-1)$. Without our proposed parameterization ($a, b = 1$), dealing with vanishing gradient is mostly possible with positive arguments, which makes the negative ones (useful for *bias shift*) hurtful for back-propagation. With the proposed parameterization, $a$ can be adjusted to increase the length $a \exp(-1)$ and allow more negative activations $h$ to counter vanishing gradients. The ratio $\frac{b}{a}$ can also be modified to ensure $w \geq \frac{b}{a}$ so that $f'(wh)w \geq 1$ for $h > 0$. Based on this analysis, the proposed parameterization gives more flexibility to the network, and the experiments in Section 4 have shown that the networks do indeed take advantage of it.

## 3.3 OPTIMIZATION

PELU is trained simultaneously with all the network parameters during back-propagation. Using the chain rule of derivation, the derivative of objective $E$ with respect to $a$ and $b$ for one layer is:

$$\frac{\partial E}{\partial a} = \sum_i \frac{\partial E}{\partial f(h_i)} \frac{\partial f(h_i)}{\partial a}, \qquad \frac{\partial E}{\partial b} = \sum_i \frac{\partial E}{\partial f(h_i)} \frac{\partial f(h_i)}{\partial b}, \tag{6}$$

where $i$ sums over all elements of the tensor on which $f$ is applied. The terms $\frac{\partial E}{\partial f(h_i)}$ are the gradients propagated from the above layers, while $\frac{\partial f(h)}{\partial a}$ and $\frac{\partial f(h)}{\partial b}$ are the gradients of $f$ with respect to $a, b$:

$$\frac{\partial f(h)}{\partial a} = \begin{cases} \frac{h}{b} & \text{if } h \geq 0 \\ \exp(h/b) - 1 & \text{if } h < 0 \end{cases}, \qquad \frac{\partial f(h)}{\partial b} = \begin{cases} -\frac{ah}{b^2} & \text{if } h \geq 0 \\ -\frac{a}{b^2}\exp(h/b) & \text{if } h < 0 \end{cases}. \tag{7}$$

To preserve the parameter positivity after the updates, we force them to always be greater than 0.1. The update rules are the following:

$$\begin{aligned} \Delta a &\leftarrow \mu \Delta a - \alpha \frac{\partial E}{\partial a} \\ a &\leftarrow \max\{a + \Delta a, 0.1\} \end{aligned}, \qquad \begin{aligned} \Delta b &\leftarrow \mu \Delta b - \alpha \frac{\partial E}{\partial b} \\ b &\leftarrow \max\{b + \Delta b, 0.1\} \end{aligned} \tag{8}$$

Table 1: Comparing PELU, ELU and ReLU with SmallNet and ResNet110 on the CIFAR-10 and CIFAR-100 tasks. The results are test error rates (in %) averaged over five tries.

| | SmallNet | | | ResNet110 | |
| | CIFAR-10 | CIFAR-100 | | CIFAR-10 | CIFAR-100 |
|---|---|---|---|---|---|
| ReLU | 13.96 | 40.51 | ReLU (He et al., 2015a) | 6.42 | 27.23 |
| ELU | 14.81 | 39.76 | ELU (Shah et al., 2016) | 5.62 | 26.55 |
| PELU | **13.54** | **38.93** | PELU (ours) | **5.37** | **25.04** |

In (8), $\mu$ is the momentum and $\alpha$ is the learning rate. When specified by the training regimes, we also use a $\ell_2$ weight decay regularization on both the weight matrices $W$ and PELU parameters. This is different than PReLU, which did not use weight decay to avoid a shape bias towards ReLU. In our case, weight decay is necessary for $a$ and $b$, otherwise the network could circumvent it for the $W$s by adjusting $a$ or $b$, a behavior that would be hurtful for training.

## 4 EXPERIMENTATIONS

In this section, we present our experiments in supervised learning on the CIFAR-10/100 and ImageNet tasks. Our goal is to show that, with the same network architecture, parameterizing ELU improves the performance. We also provide results with the ReLU activation function for reference.

### 4.1 CIFAR-10/100

As first experiment, we performed object classification on the CIFAR-10 and CIFAR-100 datasets (60,000 32x32 colored images, 10 and 100 classes respectively) (Krizhevsky et al., 2012). We trained a residual network (ResNet) with the identity function for the skip connexion and bottleneck residual mappings with shape (Conv + ACT)x2 + Conv + BN (He et al., 2015a; Shah et al., 2016). The ACT module is either PELU, ELU or BN+ReLU. We follow Facebook's Torch implementation fb.resnet.torch[1] for data augmentation and learning rate schedule, so that training is not biased towards PELU to the detriment of the other activations.

We also evaluated our proposed PELU on a smaller convolutional network. We refer to this network as SmallNet. It contains three convolutional layers followed by two fully connected layers. The convolutional layers were respectively composed of 32, 64 and 128 3x3 filters with 1x1 stride and 1x1 zero padding, each followed by ACT, 2x2 max pooling with a stride of 2x2 and dropout with probability 0.2. The fully connected layers were defined as $2048 \rightarrow 512$, followed by ACT, dropout with probability 0.5, and a final linear layer $512 \rightarrow 10$ for CIFAR-10 and $512 \rightarrow 100$ for CIFAR-100. We performed global pixel-wise mean subtraction, and used horizontal flip as data augmentation.

Table 1 presents the test error results (in %) of SmallNet and ResNet110 on both tasks, with ELU, ReLU and PELU. For SmallNet, PELU reduced the error of ELU from 14.81% to 13.54% on CIFAR-10, and from 39.76% to 38.93% on CIFAR-100, which corresponds to a relative improvement of 8.58% and 2.09% respectively. As for ResNet110, PELU reduced the error of ELU from 5.62% to 5.37% on CIFAR-10, and from 26.55% to 25.04% on CIFAR-100, which corresponds to a relative improvement of 4.45% and 5.68% respectively. These results suggest that parameterizing the ELU activation improves its performance.

It is worth noting for ResNet110 that weight decay played an important role in obtaining these performances. Preliminary experiments conducted with a weight decay of 0.0001 showed no significant improvements of PELU over ELU. We observed larger differences between the train and test set error percentages, which indicated possible *over-fitting*. By increasing the weight decay to 0.001, we obtained the performance improvements shown in Table 1. Importantly, we did not have to increase the weight decay for SmallNet. The PELU, ELU and BN+ReLU SmallNets used the same decay. Although these results suggest that residual networks with PELU activations may be more prone to over-fitting, weight decay can still be used to correctly regularize the ResNets.

---

[1] https://github.com/facebook/fb.resnet.torch

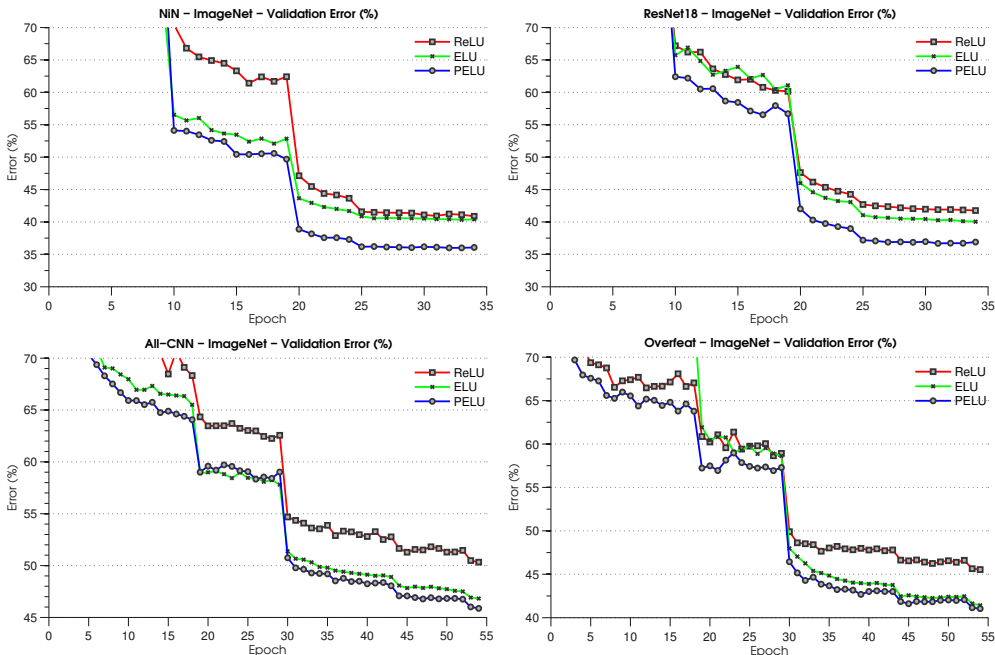

Figure 2: TOP-1 error rate progression (in %) of ResNet18, NiN, Overfeat and All-CNN on ImageNet 2012 validation set. NiN and ResNet18 (top row) used training regime #1, while All-CNN and Overfeat (bottom row) used training regime #2 (see Table 2). PELU has the lowest error rates for all networks. Regime #1 shows a greater performance gap between ELU and PELU than regime #2.

## 4.2 IMAGENET

We finally tested the proposed PELU on ImageNet 2012 task (ILSVRC2012) using four different network architectures: ResNet18 (Shah et al., 2016), Network in Network (NiN) (Lin et al., 2013), All-CNN (Springenberg et al., 2015) and Overfeat (Sermanet et al., 2013). We used either PELU, ELU or BN+ReLU for the activation module. Due to NiN's relatively complex architecture, we added BN after each max pooling layer (every three layers) for further reducing vanishing gradients. Each network was trained following Chintala's Torch implementation `imagenet-multiGPU.torch` [2] with the training regimes shown in Table 2. Regime #1 starts at a higher learning rate (1e-1) than regime #2 (1e-2), and has a larger learning rate decay of 10 compared to 2 and 5.

Figure 2 presents the TOP-1 error rate (in %) of all four networks on ImageNet 2012 validation dataset. We see from these figures that PELU consistently obtained the lowest error rates for all networks. The best result was obtained with NiN. In this case, PELU improved the error rate from 40.40% (ELU) to 36.06%, which corresponds to a relative improvement of 7.29%. Importantly, NiN obtained these improvements at little computational cost. It only added 24 additional parameters, i.e. 0.0003% increase in the number of parameters. This suggests that PELU acts on the network in a different manner than the weights and biases. Such a low number of parameters cannot significantly increase the expressive power of the network. We would not have seen such a large improvement by adding 24 additional weights to a convolutional layer with the ELU activation.

---

[2] https://github.com/soumith/imagenet-multiGPU.torch

Table 2: ImageNet training regimes.

|  | Regime #1 (ResNet18, NiN) | | | | Regime #2 (Overfeat, AllCNN) | | | | |
|---|---|---|---|---|---|---|---|---|---|
| Epoch | 1 | 10 | 20 | 25 | 1 | 19 | 30 | 44 | 53 |
| Learning Rate | 1e-1 | 1e-2 | 1e-3 | 1e-4 | 1e-2 | 5e-3 | 1e-3 | 5e-4 | 1e-4 |
| Weight Decay | 5e-4 | 5e-4 | 0 | 0 | 5e-4 | 5e-4 | 0 | 0 | 0 |

Table 3: Effect of parameter configuration using ResNets on CIFAR-10 and CIFAR-100 datasets. These results show that parameter type $(a, \frac{1}{b})$ achieves the lowest error rate (in %) in general.

| CIFAR-10 | | | | | CIFAR-100 | | | | |
|---|---|---|---|---|---|---|---|---|---|
| Depth | Parameter Type | | | | Depth | Parameter Type | | | |
| | $(a,b)$ | $(a, \frac{1}{b})$ | $(\frac{1}{a}, b)$ | $(\frac{1}{a}, \frac{1}{b})$ | | $(a,b)$ | $(a, \frac{1}{b})$ | $(\frac{1}{a}, b)$ | $(\frac{1}{a}, \frac{1}{b})$ |
| 20 | 7.71 | **7.47** | 7.64 | 7.61 | 20 | 30.17 | **30.16** | 30.26 | 30.55 |
| 32 | 6.71 | **6.41** | 6.69 | 6.53 | 32 | 28.47 | **28.01** | 28.23 | 28.56 |
| 44 | 6.52 | **5.92** | 6.51 | 6.19 | 44 | 27.70 | **27.18** | 27.24 | 27.63 |
| 56 | 6.28 | **5.65** | 6.30 | 5.75 | 56 | 26.89 | **26.33** | 26.72 | 26.40 |
| 110 | 6.22 | **5.37** | 6.09 | 5.42 | 110 | 25.69 | **25.04** | 25.59 | 25.17 |

We see from the curves in Figure 2 that the training regime has an interesting effect on the convergence of the networks. The performance of PELU is closer to the performance of ELU for regime #2, while it is significantly better than ELU for regime #2. Although we do not have a clear explanation to why this is the case, we believe that the small initial learning rate of regime #2 affects PELU optimization due to smaller gradient steps. We also see that the error rates of All-CNN and Overfeat with PELU increase by a small amount starting at epoch 44. Since ELU and ReLU do not have this error rate increase, this shows possible over-fitting for PELU. For regime #2, the error rates decrease more steadily and monotonically. Although performing more experiments would improve our understanding, these results suggest that PELU could and should be trained with larger learning rates and decays for obtaining better performance improvements.

## 5 DISCUSSION

In this section, we elaborate on other parameter configurations and perform a visual evaluation of the parameter progression throughout the training phase.

### 5.1 PARAMETER CONFIGURATION

The proposed PELU activation function (3) has two parameters $a$ and $b$, where $a$ is used with a multiplication and $b$ with a division. A priori, any of the four configurations $(a,b)$, $(a, \frac{1}{b})$, $(\frac{1}{a}, b)$ or $(\frac{1}{a}, \frac{1}{b})$ could be used as parameterization. In this section, we show experimentally that PELU with the proposed $(a, \frac{1}{b})$ configuration is the preferred choice, as it achieves the best overall results.

For evaluating the effect of parameter configuration, we trained several CNNs on the CIFAR-10 and CIFAR-100 datasets (Krizhevsky et al., 2012). We used ResNets with a depth varying from 20, 32, 44, 56 to 110. The ResNets had the identity function for the skip connexion and two different residual mappings (He et al., 2015a; Shah et al., 2016). We used a basic block Conv + PELU + Conv + BN block for depth 20, 32 and 44, and bottleneck block (Conv + PELU)x2 + Conv + BN for depth 56 and 110. We report the averaged error rate achieved over five tries.

First, we can see from the results presented in Figure 3 that the error rate reduces as the network gets deeper. This is in conformity with our intuition that deeper networks have more representative capability. Also, we can see that the proposed configuration $(a, \frac{1}{b})$ obtained the best performance overall. Configuration $(a, \frac{1}{b})$ obtained 5.37% error rate on CIFAR-10 and 25.04% error rate on CIFAR-100. We believe this improvement is due to weight decay. When using configuration $(a, \frac{1}{b})$ along with weight decay, pushing parameters $a$ and $b$ towards zero encourages PELU to be similar to ReLU. In this case, the CNN is less penalized for using ReLU and more penalized for using other parametric forms. This may help the CNN to use as much PELUs that look like ReLUs as it needs without incurring a large penalty. The experiments in section 5.2 partly supports our claim. Although the performance improvement of $(a, \frac{1}{b})$ is relatively small in comparison to the other three configurations, configuration $(a, \frac{1}{b})$ should be preferred for subsequent experiments.

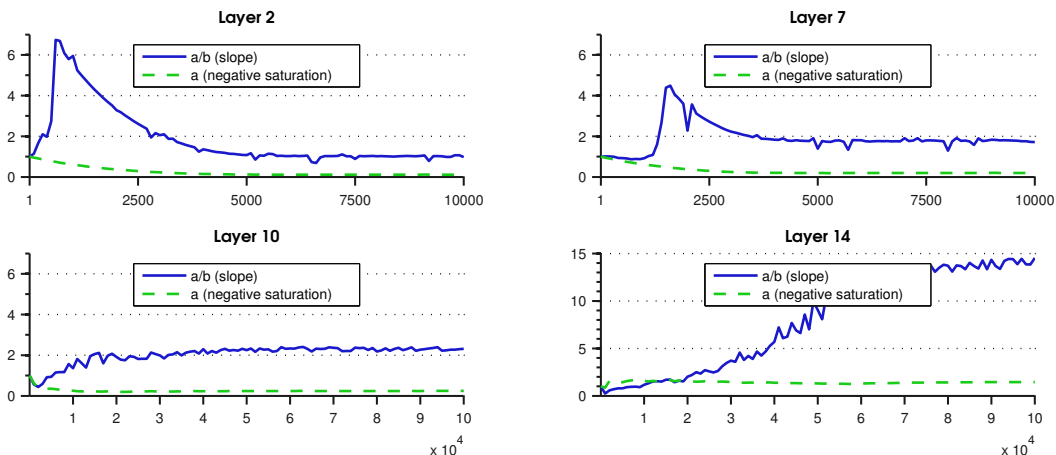

Figure 3: PELU parameter progression at layers 2, 7, 10 and 14 of Vgg trained on CIFAR-10. Interestingly, the network adopted different non-linear behaviors throughout the training phase.

## 5.2 PARAMETER PROGRESSION

We perform a visual evaluation of the non-linear behaviors adopted by a Vgg network during training (Simonyan & Zisserman, 2014). To this effect, we trained a Vgg network with PELU activations on the CIFAR-10 dataset. We performed global pixel-wise mean subtraction, and used horizontal flip as data augmentation. The trained network obtained 6.95% and 29.29% error rate on CIFAR-10 and CIFAR-100 respectively.

Figure 3 shows the progression of the slope ($\frac{a}{b}$) and the negative of the saturation point (parameter $a$) for PELU at layers 2, 7, 10 and 14. We can see different behaviors. In layer 2, the slope quickly increases to a large value (around 6) and slowly decreases to its convergence value. We observe a similar behavior for layer 7, except that the slope increases at a later iteration. Layer 2 increases at about iteration 650 while layer 7 at about iteration 1300. Moreover, in contrast to layer 2 and 7, the slope in layer 10 increases to a smaller value (around 2) and does not decrease after reaching it. Layer 14 also displays a similar behavior, but reaches a much higher value (around 15). We believe that adopting these behaviors helps early during training to disentangle redundant neurons. Since peak activations scatter the inputs more than flat ones, spreading neurons at the lower layers may allow the network to unclutter neurons activating similarly. This may help the higher layers to more easily find relevant features in the data.

The saturation point in layer 2, 7 and 10 converges in the same way to a value near zero, while in layer 14 it reaches a value near -2. This is an interesting behavior as using a negative saturation reduces bias shift. In another experiment with Vgg, we tried adding BN at different locations in the network. We saw similar convergence behaviors for the saturation point. It seems that, it this specific case, the network could counter bias shift with only the last layer, and favored sparse activations in the other layers. These results suggest that the network takes advantage of the parameterization by using different non-linear behaviors at different layers.

## 6 CONCLUSION

The activation function is a key element in Convolutional Neural Networks (CNNs). In this paper, we proposed learning a parameterization of the Exponential Linear Unit (ELU) function. Our analysis of our proposed Parametric ELU (PELU) suggests that CNNs with PELU may have more control over bias shift and vanishing gradients. We performed several supervised learning experiments and showed that networks trained with PELU consistently improved their performance over ELU. Our results suggest that the CNNs take advantage of the added flexibility provided by learning the proper activation shape. As training progresses, we have observed that the CNNs change the parametric form of their PELU both across the layers and across the epochs. In terms of possible implications of

our results, parameterizing other activation functions could be worth investigating. Functions like Softplus, Sigmoid or Tanh may prove to be successful in some cases with proper parameterizations. Other interesting avenues for future work include applying PELU to other network architectures, such as recurrent neural networks, and to other tasks, such as object detection

## ACKNOWLEDGEMENTS

We thankfully acknowledge the support of NVIDIA Corporation for providing the Tesla K80 and K20 GPUs for our experiments.

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
