# Peer review of "Parametric Exponential Linear Unit for Deep Convolutional Neural Networks"

_ICLR 2017 — rejected_

[Official Review · AnonReviewer2 · rating 6 · confidence 4 · 16 Dec 2016]
**A parameterized variant of ELU non-linearity**

Authors present a parameterized variant of ELU and show that the proposed function helps to deal with vanishing gradients in deep networks in a way better than existing non-linearities. They present both a theoretical analysis and practical validation for presented approach. 

Interesting observations on statistics of the PELU parameters are reported. Perhaps explanation for the observed evolution of parameters can help better understand the non-linearity. It is hard to evaluate the experimental validation presented given the difference in number of parameters compared to other approaches.

[Official Review · AnonReviewer3 · rating 4 · confidence 4 · 17 Dec 2016]
**Experiment design is unconvincing**

This paper proposes a modification of the ELU activation function for neural networks, by parameterizing it with 2 trainable parameters per layer. This parameter is proposed to more effectively counter vanishing gradients. 

My main concern regarding this paper is related to the authors' claims about the effectiveness of PELU. The analysis in Sections 2 and 3 discusses how PELU might improve training by combating gradient propagation issues. This by itself does not imply that improved generalization will result, only that models may be easier to train. However, the experiments all seek to demonstrate improved generalization performance.
But this could in principle be due to a better inductive bias, and have nothing to do with the optimization analysis. None of the experiments are designed to directly support the stated theoretical advantage of PELU compared to ELU in optimizing models.

In the response to the pre-review question, the authors state that the claims in Section 2 and 3.3 are meant to apply to generalization performance. I fail to see how this is true for most claims, except the flexibility claim. As the authors agree, better training may or may not lead to better out-of-sample performance. I can only agree that having flexibility can sometimes help the network adapt its inductive bias to the problem (instead of overfitting), but this is a much weaker claim compared to the mathematical justifications for improved optimization.

On selection of learning hyperparameters:
The authors state in the discussion on OpenReview that the learning rates selected were favorable to ReLU, and not PELU. However, this does not guarantee that they were not unfavorable to ELU. It raises the question: can a regime be constructed where ELU has better performance than PELU? If so, how can we draw the conclusion that PELU is better?

Overall, I am not yet convinced by the experimental setup and the match between theory and experiments in this paper.

[Official Review · AnonReviewer1 · rating 7 · confidence 5 · 28 Dec 2016]
**Parametric Exponential Linear Unit for Deep Convolutional Neural Networks**

This paper presents a new non-linear function for CNN and deep neural networks. 
The new non-linearity reports some gains on most datasets of interest, and can be used in production networks with minimal increase in computation.

[Official Review · AnonReviewer4 · rating 5 · confidence 4 · 01 Jan 2017 (modified: 02 Jan 2017)]
**Interesting and potentially very valuable - if that really works**

The paper deals with a very important issue of vanishing gradients and the quest for a perfect activation function. Proposed is an approach of learning the activation functions during the training process. I find this research very interesting, but I am concerned that the paper is a bit premature.

There is a long experimental section, but I am not sure what the conclusion is. The authors appear to be somewhat confused themselves. The amount of "maybe" "could mean", "perhaps" etc. statements in the paper is exceptionally high. For this paper to be accepted it needs a bold statement about the performance, with a solid evidence. In my opinion, that is lacking as of now. This approach is either a breakthrough or a dud, and after reading the paper I am not convinced which case it is.

The theoretical section could be made a little clearer.

Finally, how is the performance affected. The huge advantage if ReLU is in the fact that the formula is so simple and thus not costly to evaluate. How do PELU-s compare.

[Final Decision · Program Chairs · 06 Feb 2017]
**ICLR committee final decision**

The paper describes a parametric version of the exponential linear unit (ELU) activation function. The novelty of the contribution is limited, and the experimental evaluation in its current form is not convincing.